# STRUCTURED THOUGHTS FOR IMPROVED REASONING AND CONTEXT PRUNING

## ABSTRACT

Large language models (LLMs) excel at generating long chains of thought, but long reasoning traces are often verbose and memory-inefficient. In this work, we introduce **Structured Thoughts**, a framework that organizes reasoning into alternating `<try>` and `<outcome>` blocks: `<try>` captures exploratory scratch work, while `<outcome>` contains the distilled conclusion of that step. We construct a dataset of structured thoughts by segmenting reasoning traces into `<try>` blocks and prompting an LLM to summarize each step into its corresponding `<outcome>`. Fine-tuning pretrained foundation models on this reformatted data produces models that adopt the structured reasoning style, leading to performance gains of up to 8.08% on reasoning benchmarks compared to standard SFT. The explicit structure also enables context pruning: after each `<try>`/`<outcome>` pair, the `<try>` can be pruned, allowing the model to retain conclusions without keeping the full scratch work in the context. A proof-of-concept pruning implementation achieves an average of 85% memory / context savings with an 8.67% performance drop across mathematical tasks.

## 1 INTRODUCTION

Standard LLMs often fail on tasks that require multi-step logical or mathematical reasoning. Reasoning models extend LLM capabilities on such complex tasks by explicitly generating intermediate steps before arriving at a final answer. This distinction between generic LLMs and reasoning-oriented LLMs has proven to be critical for performing well on challenging domains such as competition-level mathematics, logical reasoning and programming (OpenAI, 2024; Comanici et al., 2025; Guo et al., 2025; Bercovich et al., 2025)

Transforming a pretrained model into a reasoning model requires post-training with reasoning traces. The most common first step is **supervised fine-tuning (SFT)**, where the model is trained to imitate curated chains-of-thought (CoT) that spell out intermediate steps (Wei et al., 2022). SFT is critical because it provides the model with an initial format and skill set for reasoning. Once the model has learned to reliably generate reasoning trajectories, **reinforcement learning (RL)** methods are applied to further refine them. RL teaches the model to prefer reasoning trajectories that lead to the correct answers. Recent large-scale efforts illustrate the power of this recipe. OpenAI's `o3` and `o3-mini` models, and DeepSeek's `R1` (OpenAI, 2025; Guo et al., 2025) and other models combine SFT and RL on large scale reasoning datasets, yielding state-of-the-art performance on mathematical and coding benchmarks. This indicates that reasoning-oriented post-training can substantially extend the capabilities of pretrained LLMs.

A major focus of current reasoning research is on improving the *RL stage* of post-training. These include introducing process supervision to reward intermediate steps (Lightman et al., 2023; Uesato et al., 2022; Shao et al., 2024) as well as refining the optimization objective/algorithm itself, as in recent analyses of R1-zero-style training (Liu et al.). In contrast, our work does not modify the optimization procedure or reward design. We impose a syntax in which each problem-solving step is organized into two blocks: a `<try>` block containing exploratory scratch work (mathematical derivations, verification, intermediate proofs) and an `<outcome>` block which contains the conclusion of the step. This alternation enforces a clean interface between working and result, analogous to human reasoning where we deliberate before summarizing a finding. Empirically, we find that finetuning models on such type of structured traces improves benchmark performance across

multiple foundation models. Imposing this reasoning structure further helps with the growth in sequence length, KV-cache memory, and attention cost that comes from long traces. In our structured SFT scheme, each `<outcome>` summarizes its preceding `<try>`, so the `<try>` tokens can be pruned once the conclusion is generated. We do this by masking the `<try>` blocks during training which teaches the model to rely only on outcomes for future reasoning steps. As a result, during inference, we can prune the completed `<try>` spans but retain their `<outcome>` blocks.

In summary, this paper makes two contributions:

1. We demonstrate that SFT on structured thoughts yields improvements on standard reasoning benchmarks compared to standard SFT which only contains minimal structure in the form of `<think> <answer>` tags.

2. We evaluate **structure-aware pruning**: training-time masking and inference-time pruning that discards scratch work, reducing context length and memory requirements with modest performance degradation.

## 2 RELATED WORK

**Reasoning with explicit intermediate steps.**    The recent surge in LLM reasoning performance is closely tied to techniques that elicit and exploit intermediate steps. *Chain-of-Thought* (CoT) prompting shows that models benefit from explicit, step-by-step traces (Wei et al., 2022), including the zero-shot variant that adds a simple instruction like "Let's think step by step" (Kojima et al., 2022). Decoding strategies such as *self-consistency* ensemble diverse reasoning paths and select a majority answer, further improving accuracy (Wang et al., 2022). Search-based approaches (*Tree-of-Thoughts*) widen exploration over latent "thoughts" and enable backtracking and lookahead (Yao et al., 2023). These lines of work demonstrate that supervising or decoding with intermediate steps increases reliability across math and logic based tasks.

More recently, researchers have explored directly *rewarding* intermediate steps. Process supervision methods assign rewards not just to final answers but also to individual reasoning steps. Examples include reward models trained to score intermediate solutions (Lightman et al., 2023; Uesato et al., 2022),and approaches that adaptively allocate test-time compute by rewarding useful intermediate reasoning (Snell et al., 2024). Qu et al. (2025) introduce the idea of rewarding each intermediate step if it increases the likelihood of producing the correct answer and find this improves reasoning efficiency and performance. These approaches highlight that making the intermediate structure of reasoning explicit and rewarding it can improve reasoning performance.

**Long-context and memory-efficient inference.**    A large body of literature seek to reduce attention/memory cost for long sequences by modifying attention patterns. Local/sliding-window or block-sparse attention scale linearly and combine local windows with a few global tokens or random links to preserve connectivity (Beltagy et al., 2020; Zaheer et al., 2020). Other families include hashing-based attention (Reformer) and low-rank/key-projection (Linformer). (Kitaev et al., 2020; Wang et al., 2020). Unlike these mechanisms, our approach is *data-aware*: we insert structure into the reasoning trace itself and then compress along these semantically meaningful boundaries by pruning `<try>` blocks while preserving their `<outcome>` summaries.

**PENCIL: learned reduction via call–return syntax.**    Yang et al. (2025) introduce a reduction rule that triggers when the generated sequence matches $C \ [\text{CALL}] \ T \ [\text{SEP}] \ A \ [\text{RETURN}] \Rightarrow C \ A$: the intermediate thoughts $T$ and the control tokens are deleted, and the answer $A$ is merged back into the context $C$. This enables "long thoughts with short memory" by repeatedly pruning solved subproblems, and is shown to work strongly on symbolic puzzles (e.g., nearly perfect Einstein's puzzle) with a small ($\sim$25M) transformer and a 2,048-token context.

Our work is most closely related to PENCIL but differs in important ways. (1) *Structure improves reasoning:* we show that introducing a reasoning syntax into the model *by itself* improves benchmark accuracy after supervised fine-tuning, whereas PENCIL primarily targets memory efficiency without demonstrating accuracy gains from formatting. (2) *Large-scale setting:* PENCIL evaluates on carefully curated puzzle-style tasks with a small model, while we study foundation-scale models (7B–8B) on large post-training corpora having in the order of millions of examples and on standard

reasoning benchmarks. 3) *Methodology for scalable structure and pruning:* To enable structured reasoning at scale, we (a) develop an unsupervised methodology to generate structured reasoning traces from standard reasoning datasets, (b) train models on this structured corpus that produce structured traces, and (c) introduce a second stage of *masked SFT* with custom attention masks that block access to `<try>` tokens once their `<outcome>` has been generated. These contributions are essential for scaling structural pruning beyond toy puzzles to practical LLM training.

## 3 METHOD

Our goal is to improve the accuracy and efficiency of reasoning models by introducing structure into reasoning traces and exploiting that structure for pruning. This section describes the motivation, dataset construction, model training, and pruning mechanism. We outline three stages: (i) supervised fine-tuning (SFT) on structured data, (ii) SFT with pruning-aware masking, and (iii) inference-time pruning.

### 3.1 MOTIVATION: EXPLICIT STRUCTURE FOR BETTER REASONING

For most reasoning models, the bulk of the reasoning process appears inside a single `<think>` ... `</think>` block followed by a final answer. Within this `<think>` block, models typically decompose the task into multiple sub-problems and solve them step by step. These steps often include the model utilizing various reasoning techniques such as self-reflection and verification. While useful, these techniques also result in long reasoning traces. The model produces intermediate algebra, case checks, proofs and partial conclusions for each step. We refer to these granular mathematical and logical details as the *scratch work* required to solve a sub-problem.

However, when scratch work, tentative conclusions, and final answers are left intermixed, it may obscure which tokens drive the solution, making it harder to allocate attention to the most decisive information. In addition, if scratch work and conclusions remain mixed, later steps might attend more indiscriminately to redundant or superseded details, which could increase the likelihood of errors. Consistent with this view, Liu et al. (2023) find that models tend to under-utilize information in the middle of long contexts ("lost in the middle"), and Wu et al. (2025) show that chain-of-thought length exhibits an inverted–U relationship with accuracy showing that longer traces can eventually hurt performance as redundant/irrelevant content accumulates. Additionally, retaining the entire trace in the context increases latency and memory usage, since every token must be stored in the KV cache even if only a small fraction is ultimately relevant.

Introducing an explicit structure that separates exploratory scratch work (`<try>`) from distilled conclusions (`<outcome>`) may help the model better discriminate between the reasoning process and final takeaways, reducing interference from redundant details and clarifying which information should guide subsequent steps. We make this format explicit with two tags and train the model to adopt the following reasoning format for each reasoning step:

> `<try>` exploratory reasoning `</try>`    `<outcome>` distilled conclusion `</outcome>`

Under supervised fine-tuning, the model learns to reason within `<try>` blocks and to commit the step's main finding in `<outcome>`, yielding traces where exploratory computation and distilled conclusions are clearly separated. We believe this separation makes it easy for the model to follow the logical flow of its reasoning process and help the model learn *where to attend*. Instead of processing a flat sequence where critical insights and low-level exploration are intertwined, the model is encouraged to learn that `<outcome>` blocks concentrate the key information from each reasoning step.

### 3.2 STAGE 1: STRUCTURED SUPERVISED FINE-TUNING (SFT)

**Dataset construction.** We begin with existing post-training reasoning datasets such as Llama-Nemotron Math v1.1 Bercovich et al. (2025), which contain a question and an answer with reasoning traces for mathematical problems. We preprocess these traces in two steps:

1. **Segmentation:** Our goal is to obtain a rough but useful notion of the *beginning* and *end* of each independent reasoning step. Therefore, we adopt a heuristic from prior work (Qu et al.

| Raw SFT trace | Structured trace (`<try>`/`<outcome>`) |
|---|---|
| **Question.** Find all natural numbers that can be written in the form $\frac{(a+b+c)^2}{abc}$, where $a, b, c$ are natural numbers. 

 ............................................................ 
 Wait, how about $a = 1, b = 1, c = 3$? Then $(1+1+3)^2/(1 \cdot 1 \cdot 3) = 5^2/3 = 25/3 \approx 8.333\ldots$ which is not a natural number. .................Wait, so with $a = 1, b = 4, c = 5, (1+4+5)^2/(1 \cdot 4 \cdot 5) = 10^2/20 = 100/20 = 5.$ So 5 is possible. Let's check that. $1, 4, 5$: sum is 10, squared is 100, divided by $1 \cdot 4 \cdot 5 = 20.$ Yep, $100/20 = 5.$ Perfect. How about 4? 
 ............................................................ | **Question.** Find all natural numbers that can be written ... 

 `<try>` Wait, how about $a = 1, b = 1, c = 3$? Then $(1 + 1 + 3)^2/(1 \cdot 1 \cdot 3) = 5^2/3 = 25/3 \approx 8.333\ldots$ which is not a natural number. .................Wait, so with $a = 1, b = 4, c = 5, (1 + 4 + 5)^2/(1 \cdot 4 \cdot 5) = 10^2/20 = 100/20 = 5.$ ......Yep, $100/20 = 5.$ Perfect. How about 4? `</try>` 

 `<outcome>` The values 8 and 6, are attainable, with 5 also being attainable through the combination $a = 1, b = 4, c = 5.$ `</outcome>` |

Figure 1: **Raw → Structured traces.** We convert free-form solutions into block-structured traces by (i) detecting decision cues (e.g., "Wait" "Hmm") to *segment* the answer into step-sized spans, (ii) wrapping each span as a `<try>` (scratch work), and (iii) prompting a larger instruction-tuned model to *summarize* that span into a concise `<outcome>` (the step's takeaway). The underlying problem and final answer are unchanged and the trace can be semantically pruned at `<try>`/`<outcome>` boundaries.

(2025)) to approximate step boundaries. We scan each reasoning trace for decision cues (e.g., "Wait," "Hmm," "Let me try again") and simply split the trace rules at these points. Finally, we wrap each split between `<try>` `<try>` tags. Our segmentation approach is deliberately simple and we leave more advanced segmentation techniques, such as those using using supervised models (Somasundaran et al., 2020), to future work.

2. **Summarization:** We require a brief conclusion for each `<try>` block. Ideally, we could simply instruct the reasoning model to produce a conclusion for each reasoning step. However, we find that publicly available reasoning models are poor at instruction following. Li et al. (2025) find that CoT based reasoning relates negatively with instruction following capabilities of a model as they find that CoT prevents attention to instruction-following tokens. Similarly, Fu et al. (2025) observe a similar relationship between reasoning ability and instruction following on mathematical tasks and show that instruction following capability worsens with an increase in the length of a reasoning trace.

Therefore, our solution is to obtain these summaries from an external, instruction-tuned model (Llama-70B-Instruct). For each `<try>` block, we provide (i) the original question, (ii) a short prefix of the evolving answer as local context, and (iii) the current `<try>` span, together with an explicit instruction that the goal is to extract the *main logical conclusion* of this sub-problem in one or two sentences (or a compact equation), without introducing new facts or re-deriving steps. The model's output is wrapped in an `<outcome>` tag. The actual prompt used is provided in Appendix B.

The result is a dataset where each problem consists of alternating `<try>` and `<outcome>` blocks, followed by the final solution. We fine-tune pretrained reasoning and instruction-tuned models on this structured dataset using standard cross-entropy loss. The objective encourages models to produce reasoning in the structured format. An illustration of a structured thought is provided in Figure 1.

### 3.3 STAGE 2: SFT WITH PRUNING-AWARE MASKING

Although introducing structured thought improves benchmark performance, the generated traces remain long as every token is still part of the context. Crucially, however, the `<try>`/`<outcome>` format now provides a method for exploring the *efficiency* side of reasoning. With outcomes serving as distilled summaries of each step, we want to evaluate whether models can continue reasoning effectively without retaining the full exploratory text. To investigate this, we design a training procedure that explicitly teaches models to operate under such constraints. Among possible approaches, we use a masking strategy to simulate pruning as it enforces the exact behavior we want (hide past `<try>` blocks once the paired `<outcome>` block has been generated). Moreover, it does not require any new losses and fits within our standard SFT pipeline.

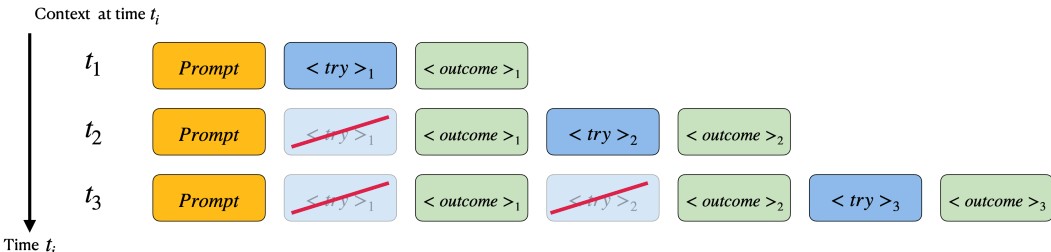

Figure 2: **Pruning procedure.** During generation the model alternates between `<try>` (blue) and `<outcome>` (green) blocks. Once an `<outcome>` is generated, its preceding `<try>` block is removed from the context. This way, only the summarized outcomes are carried forward, while earlier scratch work is discarded step by step.

**Masking strategy.** Our structured reasoning traces are compoed of alternating `<try>` and `<outcome>` blocks. Within each paired (`<try>`, `<outcome>`) segment, we have standard causal attention. However, after `<outcome>`$_i$ is produced, we masked its paired `<try>`$_i$ for all subsequent positions (including later `<try>` and `<outcome>` blocks), while keeping `<outcome>`$_i$ and all previous `<outcome>` blocks visible.

This masking simulates pruning at the context level and has two implications for computation and information flow. First, future tokens operate with a smaller effective receptive field as there is no contribution from masked `<try>` blocks. Second, information from a masked `<try>` block is not completely lost as its `<outcome>` block was computed while attending to that `<try>`, so its (key, value) content is already baked into the `<outcome>`'s representation. Since future tokens can always attend to `<outcome>` blocks, the distilled state remains available even though the original `<try>` tokens can no longer be attended to.

**Attention mask implementation.** Standard FlashAttention assumes a fixed causal mask and therefore cannot support our requirement that past `<try>` tokens become inaccessible once their corresponding `<outcome>` has been generated. To capture this behavior, we implement a rule-based masking layer using Flex Attention (Dong et al., 2024).

For each completed reasoning step, we store a triple $(c_s, c_e, o_e)$, denoting the start and end positions of the `<try>` span and the end of its corresponding `<outcome>` span. During attention, when a query token at position $q$ attends to a key token at position $k$, we apply the following rule:

$$\text{if } q \geq o_e \text{ and } c_s \leq k < c_e \quad \Rightarrow \quad \text{mask}(q, k).$$

Once the outcome of a step is present, all future tokens are prevented from attending back to the scratch tokens that produced it.

**Training procedure.** After the initial SFT on structured traces, we run a second round of SFT *on top of the structured SFT model* with pruning-aware masking enabled. The only change is the attention rule explained above.

### 3.4 STAGE 3: INFERENCE-TIME PRUNING

The final step is to implement pruning during autoregressive decoding.

**Pruning procedure.** At inference time, we implement a generate–prune–continue loop aligned to `<try>`/`<outcome>` boundaries:

1. Decode until a complete `<try>` followed by its `<outcome>` is generated.
2. Prune the `<try>` span from the context.
3. Continue decoding conditioned on the retained tokens (the prompt, tokens before the first `<try>`, and prior `<outcome>`s.

This loop runs until an answer is produced or we reach a max-length limit.

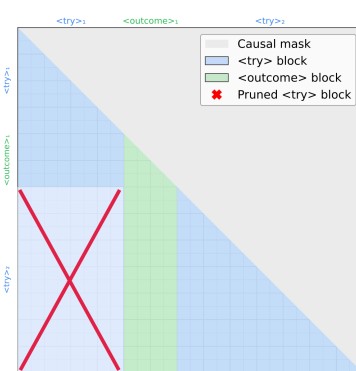

Figure 3: **Pruning-aware training mask.** The plot illustrates how attention is restricted during training. Tokens within a `<try>` block (blue) are only available until their corresponding `<outcome>` block (green) is produced. Once an `<outcome>` is generated, all subsequent queries are prevented (crossed out region) from attending back to the scratch tokens in the associated `<try>`. Future tokens can still access `<outcome>` representations, preserving distilled information. This setup teaches the model to reason with outcomes while discarding raw scratch work.

*Position encoding.* During inference we do not renumber positions after pruning. Tokens retain their original RoPE indices, so pruned spans become empty gaps in the positional timeline.

*Runtime support.* Efficient inference engines such as `vLLM` (Kwon et al., 2023) do not expose the fine-grained control needed for selective eviction and restarts at `<try>`/`<outcome>` boundaries. We therefore implement pruning using a custom decoding loop in HuggingFace (Wolf et al., 2019), which allows direct manipulation of cache states.

## 4 EVALUATION

We evaluate Structured Reasoning across two reasoning models on multiple mathematical benchmark tasks. Our experiments and analysis addresses three questions: (i) Does structured SFT improve reasoning ability compared to unstructured SFT? (ii) Is it possible to prune the context without catastrophic performance degradation? (iii) What is the overhead introduced by our structured SFT technique and gains in memory/compute from subsequent pruning.

### 4.1 DATASETS

We build on the `llama-nemotron-math-v1.1` corpus (Bercovich et al., 2025), a large-scale collection of math reasoning traces. The full dataset contains roughly 2M examples covering a wide range of problem types. For our experiments, we select the first 1M examples from this dataset. We use this dataset in two ways:

1. **Baseline SFT:** supervised fine-tuning on the raw `math v1.1` dataset, where each example consists of a question and a free-form reasoning trace.

2. **Structured SFT:** supervised fine-tuning on our reprocessed version of `math v1.1`, where each trace is reformatted into alternating `<try>` and `<outcome>` blocks. The `<try>` spans are obtained by segmenting raw reasoning traces at decision tokens (e.g., "Wait," "Hmm"), while the corresponding `<outcome>` blocks are generated by prompting a larger instruction-tuned model (Llama-70B-Instruct) to summarize each `<try>` into a concise logical conclusion. This produces scratch–summary units without changing the underlying problems or answers.

Structured SFT does not introduce new problems or answers. It reuses the exact same 1M examples as the Baseline SFT, differing only in wrapping the reasoning steps with `<try>` tags and the addition of `<outcome>` blocks

### 4.2 MODELS

We experiment with two pretrained reasoning models:

1. **Llama-Nemotron-8B** (Bercovich et al., 2025), part of the Llama-Nemotron family of reasoning models. These models incorporate post-training stages specialized for mathematical reasoning. Nemotron models are trained with a mixture of reasoning-oriented datasets and optimized with supervised CoT traces and reinforcement learning.

2. **Qwen2.5-7B-Instruct (s1)** (Hui et al., 2024) is part of the Qwen2.5 family of models. The 7B-Instruct variant is aligned via instruction tuning and further adapted, using SFT, on the `s1K-1.1` dataset (Muennighoff et al., 2025), which contains reasoning-style traces. We use the checkpoint released by Muennighoff et al. (2025) for our experiments.

### 4.3 TRAINING DETAILS

We conduct all experiments on a distributed setup of 8 nodes, each equiped with 8 NVIDIA A100 40GB GPUs connected via EFA. Our training pipeline uses veRL Sheng et al. (2025) and LlamaFactory Zheng et al. (2024). For the main experiments, we fine-tune each model for 2 epochs with a learning rate of $1 \times 10^{-4}$. We set the maximum context length to 25,000 tokens, enabling the models to handle long reasoning traces without truncation. To further improve training efficiency, we apply sequence packing so that multiple shorter samples are concatenated into a single sequence, maximizing GPU utilization.

### 4.4 STRUCTURED SFT IMPROVES REASONING PERFORMANCE

We first compare Baseline SFT (training on raw `math-v1.1` traces) against Structured SFT (training on our reformatted traces with `<try>`/`<outcome>` blocks). Evaluation is conducted on seven math reasoning benchmarks: **Math500**, **MinervaMath**, **AMC23**, **AIME24**, **TheoremQA**, **OlympiadBench**, and **GSM8K** (Hendrycks et al., 2021; Lewkowycz et al., 2022; AMC; AIM; Chen et al., 2023; He et al., 2024; Cobbe et al., 2021). Across tasks, Structured SFT consistently outperforms the raw baseline. On **Llama-Nemotron-8B**, Structured SFT yields a relative average gain of **+3.66%** over the baseline SFT experiments. On **Qwen2.5-7B-Instruct**, Structured SFT improves performance by an average of **+8.08%**. This demonstrates that structured reasoning provides measurable gains in accuracy. Table 1 reports the full benchmark-level breakdown.

| Benchmark | Llama-Nemotron-8B | | | Qwen2.5-7B-Instruct (s1) | | |
| | Score | | | Score | | |
| | Baseline SFT | Structured SFT | Diff (%) | Baseline SFT | Structured SFT | Diff (%) |
| --- | --- | --- | --- | --- | --- | --- |
| MATH-500 | 90.2 | 93.2 | +3.32% | 88.0 | 88.4 | +0.45% |
| Minerva-Math | 48.2 | 53.3 | +10.60% | 45.1 | 46.3 | +2.66% |
| GSM8K | 90.4 | 91.3 | +1.00% | 92.9 | 91.3 | -1.72% |
| OlympiadBench | 60.9 | 61.9 | +1.64% | 55.0 | 57.8 | +5.09% |
| AMC23 | 92.5 | 97.5 | +5.41% | 75.0 | 87.5 | +16.07% |
| AIME24 | 60.0 | 63.3 | +5.50 % | 33.3 | 53.3 | +60.06% |
| TheoremQA | 55.0 | 54.9 | -0.18% | 55.8 | 56.5 | +1.25% |
| Average | 71.0 | 73.6 | +3.66% | 63.58 | 68.7 | +8.08% |

Table 1: **Structured SFT improves reasoning accuracy across benchmarks.** Performance of Baseline SFT (raw traces) vs. Structured SFT (`<try>`/`<outcome>` traces) on seven math reasoning benchmarks. Structured SFT yields consistent gains: +3.66% relative improvement for Llama-Nemotron-8B, and +8.08% for Qwen2.5-7B-Instruct.

### 4.5 PRUNING EXPERIMENTS

Next, we evaluate pruning at inference time using the Llama-Nemotron-8B model. Here, the generation loop removes `<try>` tokens as soon as their corresponding `<outcome>` is produced, retaining only the compressed reasoning summaries in context. This reduces the effective sequence length and KV cache footprint, with the tradeoff that the model no longer has access to its raw scratch work.

Table 2 reports performance across math reasoning benchmarks. Pruning leads to an average relative degradation of $8.76\%$. Importantly, the model remains competitive overall, with performance levels that demonstrate pruning is feasible as a proof-of-concept. The observed performance drop can be explained by several factors. First, the segmentation technique used to identify independent reasoning steps is heuristic and can misalign with the actual boundary of the step. While this has little impact during structured SFT as the full scratch work remains visible, pruning makes the model fully dependent on the outcome blocks. If a segmentation boundary cut occurs in the middle of a step and then the preceding `<try>` block is removed, the model is left with an incomplete summary and cannot properly continue the reasoning chain. Second, the informativeness of outcome blocks

is constrained by the summarization model (Llama-70B in our case). If the summarization model misses or misinterprets the logical conclusion of a `<try>` block, then the true conclusion of that step is effectively lost, and after pruning the model has no way to recover it.

Additionally, while our results highlight that structured traces allow the model to sustain reasoning even without retaining the full scratch space, our current pruning methodology is not efficient enough for deployment. Our HuggingFace-based inference loop is significantly slower than optimized engines such as `vLLM`, and pruning adds further overhead.

## 4.6 Overhead and Efficiency Analysis

Introducing structured thoughts increases sequence length because each reasoning step now includes an additional `<outcome>` block. On our structured SFT dataset, each record contains on average 7.34 `<try>` blocks. The typical`<try>` block spans 688.5 tokens, while the paired `<outcome>` summary averages 103.2 tokens. This yields a ratio of outcome-to-try tokens of 0.15. This yields a compression ratio of almost 85% overall. If we also prune the previous `<try>` blocks while keeping the `<outcome>` before final answer, we shrink both the memory footprint (KV-cache size) and attention FLOPs during inference.

| Llama-Nemotron-8B | Benchmark | | | | | | | |
|---|---|---|---|---|---|---|---|---|
| | MATH-500 | Minerva-Math | GSM8K | OlympiadBench | AMC23 | AIME24 | TheoremQA | Average |
| Structured SFT | 93.2 | 53.3 | 91.3 | 61.9 | 97.5 | 63.3 | 54.9 | 73.6 |
| Structured SFT + Pruning | 89.6 | 50.7 | 86.0 | 55.9 | 87.5 | 46.7 | 53.8 | 67.2 |
| Diff (%) | -3.86% | -4.87% | -5.80% | -9.69% | -10.25% | -26.22% | -2.00% | -8.76% |

Table 2: **Structured SFT with pruning trades accuracy for efficiency.** Benchmark performance of Llama-Nemotron-8B trained with Structured SFT, evaluated with and without pruning. Pruning discards `<try>` tokens once their `<outcome>` is produced, reducing context length and KV-cache size. While accuracy drops by an average of $8.76\%$, the results show that the model remains competitive across benchmarks, validating pruning as a viable proof-of-concept for memory-efficient reasoning.

## 4.7 Why does Structured SFT help?

A natural concern is that our `<outcome>` additions might (i) inject new knowledge, or (ii) help merely by lengthening the trace and thus increasing the model's effective compute/memory. We offer some reasoning behind why simply adding more compute does not always scale performance and also design an ablation to isolate both factors.

**Does a longer context by itself help reasoning?** Prior work shows that simply providing (or training for) longer contexts does not automatically improve reasoning, and can even degrade effective usage of evidence. For example, Liu et al. (2023) report that LMs over-attend to the beginning/end of long prompts and under-utilize information in the middle.

Wu et al. (2025) provide evidence that longer chains-of-thought are not always better. They show that accuracy follows an inverted–U curve as the number of reasoning steps grows. Performance improves at first, then degrades as chains become too long. They further show two scaling properties: (i) *larger models* reach peak accuracy with *shorter* chains, and (ii) harder tasks benefit from longer CoT chains. Taken together, this work cautions that raw token count is an unreliable proxy for better performance on reasoning tasks.

**Ablation: no information in `<outcome>` (compute matched variant).** To test whether our gains come from extra tokens (i.e., more internal compute) rather than structured reasoning traces, we create a Compute-Matched variant. Starting from the Structured SFT data, we replace every token inside each `<outcome>` block with a single MASK symbol repeated to match the original length (thus preserving sequence length and training FLOPs), while keeping the `<try>` content unchanged. No semantic summary remains; only the token budget does. We fine-tune **Llama-Nemotron-8B** on this masked corpus for the same number of steps and hyperparameters as Structured SFT.

| Llama-Nemotron-8B | Benchmark | | | | | | | |
|---|---|---|---|---|---|---|---|---|
| | MATH-500 | Minerva-Math | GSM8K | OlympiadBench | AMC23 | AIME24 | TheoremQA | Average |
| Baseline SFT | 90.2 | 48.2 | 90.4 | 60.9 | 92.5 | 60.0 | 55.0 | 71.0 |
| **Structured SFT** | 93.2 | 53.3 | 91.3 | 61.9 | 97.5 | 63.3 | 54.9 | 73.6 |
| **Compute-Matched SFT** | 93.2 | 52.2 | 90.0 | 62.4 | 95.0 | 56.7 | 56.1 | 72.2 |
| **Performance Gain vs. Baseline (%)** | | | | | | | | |
| **Structured SFT** | +3.32 | +10.60 | +1.00 | +1.64 | +5.41 | +5.50 | -0.18 | +3.66 |
| **Compute-Matched SFT** | +3.32 | +8.30 | -0.44 | +2.46 | +2.70 | -5.50 | +2.00 | +1.69 |

Table 3: **Structured SFT vs. compute-matched control on Llama-Nemotron-8B.** We compare *Baseline SFT*, *Structured SFT*, and a *Compute-Matched* (CM) variant in which `<outcome>` tokens are replaced by mask tokens to preserve sequence length and training compute while removing summary content. While both Structured SFT and CM exceed the baseline, Structured SFT attains a larger average gain (+3.66% vs. +1.69%).

**Findings.** Table 3 summarizes the results for our compute-matched ablation. Relative to the Raw SFT baseline, *Structured SFT* yields an average gain of **3.66%**, whereas the *Compute-Matched* variant (same sequence length but contains mask tokens inside the `<outcome>` blocks) improves by **1.69%**. Since the compute-matched setting preserves token budget and training FLOPs, these results suggest that increased sequence length alone does not fully explain the improvements and that the distilled conclusions in `<outcome>` blocks may be providing additional, structure-specific benefits.

## 5 LIMITATIONS & FUTURE WORK

Our study has several limitations. First, inference-time pruning results are available only for a single model, **Llama-Nemotron-8B**. Limited compute resources prevented us from running systematic pruning experiments on other models. Second, even for the models where we tested pruning, our current runtime approach is not practical. Since vLLM does not support dynamic KV-cache eviction, we fall back to HuggingFace Transformers with which introduces a large overhead, making pruning quite slow.

Additionally, while structured SFT improves performance, pruning currently incurs a non-trivial accuracy drop. On Llama-Nemotron-8B, pruning leads to an average 8.76% relative degradation compared to the unpruned structured model. To mitigate this degradation under pruning, an additional RL phase could be introduced. Here, the model would learn to compress its own scratch work into minimal but sufficient outcomes, with pruning performed during RL training itself. This could reduce reliance on heuristic segmentation and allow the model to internalize pruning decisions.

## 6 CONCLUSION

We introduced Structured Reasoning, a method to organize reasoning traces into alternating `<try>` (scratch work) and `<outcome>` (distilled conclusion) blocks. By fine-tuning models on reformatted datasets, we showed that structured supervision improves reasoning accuracy across multiple math benchmarks (relative improvement of 3.66% for Llama-Nemotron-8B and 8.08% for Qwen2.5-7B-Instruct (s1)). Building on this structure, we further explored *pruning*, where scratch work for reasoning steps is masked after their outcomes are produced. Our experiments demonstrate that pruning is feasible and significantly reduces context length. Pruning yields 85% savings in context and memory compared to the structured reasoning variant although this comes at the cost of an average performance degradation of 8.76% on benchmark scores.

## 7 ETHICS

This work focuses on methods for improving the reasoning efficiency and structure of large language models through supervised fine-tuning and pruning. We do not introduce new datasets and the models and datasets used in this paper are already publicly publicly available. As such, this work

does not raise novel ethical or societal risks beyond those already associated with large language models.

## 8 REPRODUCIBILITY

We have made significant efforts to ensure the reproducibility of our results.

- **Dataset construction:** Section 3.2 details the preprocessing pipeline, including segmentation, summarization, and formatting of the structured traces.
- **Training setup:** Section 4.3 describes our hardware environment, frameworks, hyperparameters, and context length settings. We use publicly available training frameworks highlighted in Section 4.3
- **Masking and pruning:** Section 3.3 provides technical details of our masking implementation and inference-time pruning procedure.
- **Evaluation:** Section 4 outlines SFT datasets, benchmark datasets and experimental details.
- **Code Release:** Upon acceptance, we plan on open-sourcing our entire codebase.

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

## A  USE OF LLMs

We used LLMs for plotting code, latex formatting, finding relevant work and writing edits.

## B  SUMMARIZATION PROMPT

We use the following instruction prompt to generate summaries for each `try` block.

## ORIGINAL TEXT STRING

> **Key Idea**
>
> You are an expert in analyzing and distilling complex reasoning processes. You will receive **ONE** segment of a larger, multi-step reasoning trace. This segment is enclosed within $<try></try>$ tags.
>
> Your task is to process **ONLY THIS SINGLE** $<try>$ block and identify and provide its **most important conclusion or core logical outcome**. This outcome **MUST include any relevant mathematical equations, numerical results, or specific values** if they represent the main finding of the block.
>
> **CRITICAL RULES FOR YOUR OUTPUT:**
> 1. Your summary **MUST** directly state the outcome of *this specific block*. It should be concise, but may span multiple sentences if a mathematical formulation or detailed numerical result is the core outcome.
> 2. Your summary **MUST NOT** contain introductory phrases like "The main finding is...", "This block concludes...", "The result of this call is...", etc. Go straight to the point.
> 3. Your summary **MUST** be enclosed within $<outcome></outcome>$ tags.
> 4. Your entire output should be **ONLY** this single $<outcome></outcome>$ block, with absolutely no other text, comments, or conversational filler.
> 5. **If the main outcome is a mathematical expression or numerical result, represent it clearly within the summary, using LaTeX formatting (e.g., $E = mc^2$) for equations.**

