# OpenReview forum: "Structured Thoughts For Improved Reasoning And Context Pruning"
_ICLR.cc/2026/Conference — ICLR 2026 Conference Withdrawn Submission_

### Official Review · Reviewer_BUy7 · 2025-10-20

**Soundness:** 2
**Presentation:** 1
**Contribution:** 1
**Rating:** 2
**Confidence:** 5

**Summary:**

This paper proposes Structured Thoughts, enforcing an explicit step-wise structure that separates exploratory reasoning (<try>) from distilled conclusions (<outcome>) addressing the verbosity and memory inefficiency of long CoT reasoning. Specifically, author introduces a structured supervised fine-tuning scheme with pruning-aware masking and an inference-time pruning mechanism that removes <try> blocks after their outcomes are generated. Experiments on Llama-Nemotron-8B and Qwen2.5-7B-Instruct demonstrate accuracy gains from structured fine-tuning compared to naive SFT, and also shows memory inefficiency gain by pruning the previous generated CoT.

**Strengths:**

The paper presents a method that is conceptually coherent, proposing both a structured training (fine-tuning) approach and an inference-time pruning strategy.

**Weaknesses:**

The writing could be improved. For example, the introduction jumps straight into describing the method without providing sufficient motivation. The differences from existing related work are also discussed only at a very high level. The author should further explain why structured thought is necessary (e.g., due to memory efficiency issues and the often verbose nature of reasoning as explained in the abstract).

The related work section is insufficiently reviewed. For instance, there are existing studies such as [1] that propose methods to reduce the thinking process during inference through summarization.

The paper also fails to compare against baselines that emphasize efficiency. For example, it does not include comparisons with prior works like PENCIL [2], or [1,3].

We are observing a clear trend toward scaling via RL rather than SFT these days (indeed, several recent studies argue that SFT leads to stronger memorization compared to RL). However, this method can only be applied to SFT, which limits its generality. RL baselies such as PPO [4] or GRPO [5] results should be included, and it remains unclear whether the proposed method could extend to RL settings.

The experiments are limited. Only two models were trained on a single dataset. The evaluation should include a broader range of models and datasets.

The efficiency analysis is insufficient. It is also unclear how efficieny the approach actually is. The paper should report, for each dataset, the exact reduction in tokens compared to baselines, as well as the precise number of tokens saved. Furthermore, such attention masking might make it difficult to leverage techniques like FlashAttention, so the real decoding time should be reported.

Overall, the paper’s claims are weak and there are many aspects that could be improved. I would recommend revising the draft based on the current reviews and targeting the next venue.

Reference\
[1] Think Clearly: Improving Reasoning via Redundant Token Pruning\
[2] PENCIL: Long Thoughts with Short Memory\
[3] Beyond the Last Answer: Your Reasoning Trace Uncovers More than You Think\
[4] Proximal Policy Optimization Algorithms\
[5] DeepSeekMath: Pushing the Limits of Mathematical Reasoning in Open Language Models

**Questions:**

See the weakness above.

---

### Official Review · Reviewer_NDfE · 2025-10-29

**Soundness:** 3
**Presentation:** 3
**Contribution:** 3
**Rating:** 8
**Confidence:** 4

**Summary:**

The authors present an interesting way to condense the reasoning process of an LM. Specifically, they manage to construct training data in the form of [object Object] / [object Object], where they separate the reasoning versus a brief summary of the reasoning step. The authors then train an LM based on such data. But for the future steps, they mask out the existing steps' try block and only maintain the outcome (summary) of the existing steps. This reduces the number of tokens in the reasoning process (in the actual inference time as they can omit these intermediate reasoning block while only maintaining the brief summaries for each step). The proposed method improves the baseline models on different benchmarks.

**Strengths:**

- I like the idea of partially omitting the reasoning step while maintaining the summarized version of the outcome for each step. Very interesting and clean idea.
- The authors have conducted experiments using two baseline LLMs, and demonstrate their results on a wide range of benchmarks.

**Weaknesses:**

- In order to employ this method, the authors need to craft the data in certain format, which may hinder the generalizability of the method inherently.
- In terms of data construction, it would be nice to see what happens if we use different sets of LLMs to produce the data, just to demonstrate the generalizability of the method as different LLMs may generate the reasoning trace in different fashions.
- Typos: line 184: `using using supervised models`.

**Questions:**

- How does the keyword selection influence the model's performance?
- Can there be other structures to condense the LLMs' reasoning process?
- What if there are some dependencies across different steps, can the summarized outputs capture those dependencies? Or in general, how well do the summaries of each step capture what's really going on within the step?
- What are the tolerance of the LLMs' reasoning correctness with respect to each of the summaries' correctness? As many prior studies have shown that we may flip parts of CoT while still maintaining the answer correctness.

---

### Official Review · Reviewer_drjy · 2025-10-29

**Soundness:** 2
**Presentation:** 2
**Contribution:** 1
**Rating:** 2
**Confidence:** 3

**Summary:**

This paper proposes a reasoning-training method based on structured modeling.
Specifically, the authors explicitly divide the reasoning step and summarization step using special tokens (e.g., \<try>, \<outcome>).
Experimental results show that the proposed training method outperforms naïve SFT and can be leveraged for memory-efficient inference.

**Strengths:**

- The paper is easy to follow and clearly written.
- The proposed method is simple and conceptually clear.

**Weaknesses:**

**[W1] Novelty of method.**
The idea that summarization improves reasoning performance has already been explored in prior works [1, 2].
This reduces the novelty of the main contribution.

[1] Yan et al., INFTYTHINK: BREAKING THE LENGTH LIMITS OFLONG-CONTEXT REASONING IN LARGE LANGUAGEMODELS, Arxiv 2025

[2] Choi et al., Think Clearly: Improving Reasoning via Redundant Token Pruning, EMNLP 2025 (Findings)

**[W2] Practicality of  pruning**
The authors claim that their structured modeling enables pruning to reduce inference overheads.
However, as shown in Table 2, there is a significant performance drop across datasets (e.g., 26.22% drop on AIME24).
It is unclear whether this pruning has practical benefits, especially since KV cache compression techniques [2, 3] already achieve substantial memory savings for reasoning models.

[2] Choi et al., Think Clearly: Improving Reasoning via Redundant Token Pruning, EMNLP 2025 (Findings)

[3] R-KV: Redundancy-aware KV Cache Compression for Reasoning Models, NeurIPS 2025

**[W3] Limited baselines**
In the main results, only naïve SFT is used as a baseline. More comparisons with recent reasoning-oriented training methods are needed to better contextualize the proposed method’s effectiveness.

**Questions:**

**[Q1] Comparison to naive approach, other works.**
During reasoning, one could directly generate a short summarization at each step and append it to the ongoing reasoning process — a training-free approach.
Have the authors compared this with their structured training method?
Also, comparison with existing summarization-based training such as INFTYTHINK [1] is required.

[1] Yan et al., INFTYTHINK: BREAKING THE LENGTH LIMITS OFLONG-CONTEXT REASONING IN LARGE LANGUAGEMODELS, Arxiv 2025

**[Q2] Test-time scaling**
Structured modeling might reduce reasoning diversity and test-time scaling ability of the model. Can the proposed structured modeling preserve the scaling benefits of the original reasoning models?

**[Q3] Reasoning length**
Summarization naturally removes redundant reasoning steps, potentially reducing reasoning length.
Does the proposed method alleviate the overthinking phenomenon observed in long-chain reasoning models?

---

### Official Review · Reviewer_BowT · 2025-10-30

**Soundness:** 2
**Presentation:** 3
**Contribution:** 2
**Rating:** 2
**Confidence:** 4

**Summary:**

This paper proposes a framework to make reasoning more efficient by reformatting chain-of-thought traces into alternating <try> and <outcome> blocks. <try> blocks contain main unsctructured reasoning, and <outcome> blocks contain summaries of key conclusions. The authors create a training dataset by automatically segmenting existing reasoning traces at decision points and using a larger model to generate the summary blocks. There are two main contributions of this method: 1. improved reasoning (via structure) and 2. more efficient reasoning via pruning the structure.

**Strengths:**

- Paper is well written and motivated nicely
- The novelty is nice, and the idea of introducing structure with the goal of pruning is solid
- Method simple to follow
- The proposed method being able to simultaneously improve reasoning or prune is a positive (although, it doesn't show both).

**Weaknesses:**

- The main weakness is a lack of comparison with existing distillation or pruning methods. Ultimately, this is distilling information from the larger 70B model, and there should be simple distillation comparisons such as knowledge distillation (Hinton et al 2015). It's unclear whether the gains are coming from distillation or from the structure. There also aren't solid comparisons with methods that seek to reduce overthinking such as length penalty, thinkprune, etc. (or other methods in survey paper Stop Overthinking: A Survey on Efficient Reasoning for Large Language Models)
- I would argue that the performance tradeoff in pruning is quite large. I'm not sure how widely adopted a method that reduces scores by 10% would be, even at the efficiency gains.
- For the reasoning improvement experiments, the length is increasing, which could be the major contributor to the gains (similar to distillation argument). It's unclear from the ablations where the gains are really coming from.
- Pruning is done on your SFT'd model, so it could be that this just helps pruning in general. Pruning experiments would be better off if you could also show that it works on general structured generation from the original model.

**Questions:**

- My biggest concern is it's unclear where the gains are coming from. I think the paper needs more analysis of the distillation effect. Why not try the same size model that you're training to do the summaries?
- I think this should be shown to work on other models. E.g. would it work on a thinking model (e.g. qwen3)? The method of choosing where thinking starts/ends (sec 3.2) seems to have a big effect on the final outcome. Reasoning models already give explicit think/non-think locations.

---

### Note · Authors · 2025-11-13

I have read and agree with the venue's withdrawal policy on behalf of myself and my co-authors.